# A Metagenomic Investigation of the Viruses Associated with Shiraz Disease in Australia

**DOI:** 10.3390/v15030774

**Published:** 2023-03-17

**Authors:** Qi Wu, Nuredin Habili, Wycliff M. Kinoti, Stephen D. Tyerman, Amy Rinaldo, Linda Zheng, Fiona E. Constable

**Affiliations:** 1School of Agriculture, Food and Wine, University of Adelaide, Waite Precinct, PMB 1, Glen Osmond, SA 5064, Australia; 2Australian Wine Research Institute, Wine Innovation Central Building, Hartley Grove crn Paratoo Road, Urrbrae, SA 5064, Australia; 3Agriculture Victoria Research, Department of Energy, Environment and Climate Action, AgriBio, Centre for AgriBioscience, 5 Ring Road, Bundoora, VIC 3083, Australia; 4School of Applied Systems Biology, La Trobe University, Bundoora, VIC 3086, Australia

**Keywords:** Shiraz disease, Australia, grapevine virus A, grapevine leafroll-associated virus 3, metagenomic sequencing, grapevine leafroll-associated virus 4, phylogenetic groups, dsRNA

## Abstract

Shiraz disease (SD) is an economically important virus-associated disease that can significantly reduce yield in sensitive grapevine varieties and has so far only been reported in South Africa and Australia. In this study, RT-PCR and metagenomic high-throughput sequencing was used to study the virome of symptomatic and asymptomatic grapevines within vineyards affected by SD and located in South Australia. Results showed that grapevine virus A (GVA) phylogroup II variants were strongly associated with SD symptoms in Shiraz grapevines that also had mixed infections of viruses including combinations of grapevine leafroll-associated virus 3 (GLRaV-3) and grapevine leafroll-associated virus 4 strains 5, 6 and 9 (GLRaV-4/5, GLRaV-4/6, GLRaV-4/9). GVA phylogroup III variants, on the other hand, were present in both symptomatic and asymptomatic grapevines, suggesting no or decreased virulence of these strains. Similarly, only GVA phylogroup I variants were found in heritage Shiraz grapevines affected by mild leafroll disease, along with GLRaV-1, suggesting this phylogroup may not be associated with SD.

## 1. Introduction

Shiraz disease (SD) and grapevine leafroll disease (LRD) are the two major viral diseases that pose a serious threat to Australia’s AUD 45.5 billion dollar grape and wine industry [1,2,3]. LRD is associated with grapevine leafroll-associated viruses (GLRaVs) of the genera *Ampelovirus*, *Closterovirus* and *Velarivirus* within the family *Closteroviridae* [4]. Grapevine leafroll-associated virus 3 (GLRaV-3) is considered to be the most important GLRaV species due to its global prevalence, the severity of associated disease and impact on production [5]. In red berry varieties, LRD symptoms include downwards rolling of red leaves with green veins, which commences at veraison [5]. Studies have shown that sensitive grapevine varieties infected with GLRaV-3 can have a significant reduction in vigour, yield, juice sugar and berry anthocyanins [6,7,8]. Conversely, white berry varieties are often asymptomatic where vigour, fruit quality and yield may not be affected [5,9].

SD is only known to occur in South Africa and Australia and the disease is associated with significant yield losses and grapevine decline [1,2,10,11]. SD was first reported from South Africa, affecting Shiraz and Merlot grapevines grafted onto 101-14 rootstock [12]. The symptoms of SD include uneven lignification of canes (ULC) and purple leaves that remain longer on affected grapevines at the end of the growing season as compared to the asymptomatic grapevines [12]. SD was also described in Australia as early as 2001 [13], with affected grapevines showing late bud burst, restricted spring growth (RSG), leaf reddening and ULC in autumn [14,15]. In both countries, SD is associated with grapevine virus A (GVA), and in South Africa, a strong association was observed between GVA variants in phylogenetic group II (GVA^II^) [11]. The association between specific GVA variants and SD in Australia has not been explored in depth, although GVA^II^ variants were found in seven SD-affected plants of cv. Shiraz and Merlot [10]. Symptoms of SD vary between grapevine varieties. Red berry varieties including Shiraz, Merlot, Malbec, Gamay, Ruby Cabernet and Sumoll express typical SD symptoms when infected with GVA, whereas Cabernet Sauvignon, Grenache, Nero d’Avola, and most white berry varieties and rootstocks are consistently asymptomatic when infected with GVA [1,2,16]. 

GVA has five open reading frames (ORFs): ORF1 encodes a 194 kDa polypeptide functioning as an RNA-dependent RNA polymerase (RdRp), ORF2 encodes a 19 kDa polypeptide with unknown function, ORF3 encodes a 31 kDa movement protein (MP), ORF4 encodes a 21.5 kDa coat protein (CP) and ORF 5 encodes a 10 kDa polypeptide putatively involved in RNA-binding (RB) functions by suppressing RNA-silencing responses [17,18]. Symptom severity in *Nicotiana benthamiana* inoculated with GVA depended upon the eighth amino acid (aa) from the N-terminus of the RNA-binding gene [19,20]. However, the association between specific genes or changes in specific nucleotides or amino acids and symptoms of SD within the grapevine is not known. 

In this study, we use endpoint reverse transcription polymerase chain reaction (RT-PCR) and metagenomic high-throughput sequencing (Meta-HTS) to investigate the association between SD and GVA, its variants and other viruses in affected Australian grapevines.

## 2. Materials and Methods

### 2.1. Vineyard Locations and Grapevine Selection

Two commercial vineyards with SD-affected grapevines cv Shiraz, at Langhorne Creek (LC) and Willunga (WIL; blocks 1 and 2 Appendix A) in South Australia (SA), and one commercial vineyard with mild LRD-affected heritage Shiraz at Barossa Valley (BV), SA, were chosen for this study. A total of 116 grapevines, including SD- or LRD-affected and asymptomatic grapevines, were selected to investigate the virome and association of viruses and their strains to disease. Samples were collected over three years, in May to July of 2018–2019, March 2019 and August 2020 for RT-PCR testing (Table 1 and Appendix A). Forty-one samples, which represent distinct symptom types at each location, were selected from the 116 samples for Meta-HTS. The SD grapevines were selected based on the presence of both the RSG symptoms in the early growing stages around EL stage 15 (8 leaves separated [21]), and leaf reddening observed [1] (Figure 1a,b) from EL stage 35 (veraison) to leaf-fall. The LRD grapevines were selected based on field observations of red leaves with green veins around EL stage 38 (harvest). Mild LRD symptoms refer to pinkish reddening leaves (BV) or slight red leaves without rolling (Figure 1c). Asymptomatic grapevines did not show RSG, SD or LRD at the time of selection and appeared healthy (Figure 1d). Symptom types were identified twice by field observation at EL stage 15 for RSG and at EL stage 38 for leaf reddening at each study site, from 2018 to 2020.

To investigate the potential for spread between WIL and adjoining blocks of grapevines, four grapevines (Cabw1, 2, 11 and 12) from an adjoining block of Cabernet Sauvignon (clone SA125) and a grapevine (Merlot1) from an abandoned neighbouring Merlot block were selected for virome analysis (Appendix A, Table 1). Merlot1 showed typical SD symptoms and Cabernet Sauvignon showed LRD symptoms at EL-38 (Table 1). 

Two asymptomatic own-rooted Shiraz (clone BVRC12) grapevines (Shiraz_OR_P5 and Shiraz_OR_P6) and two Cabernet Sauvignon (clone SA125) grapevines (CabSA125_R3V30 and CabSA125_R3V44) with mild LRD symptoms located in the Coombe’s research vineyard (CV), Waite Campus, University of Adelaide, SA, were selected as controls for diseased grapevines’ virome comparison (Table 1).

### 2.2. Sample Collection 

The sampling time and tissue type for the RT-PCR and HTS experiments are listed in Appendix A. Petioles (5 per grapevine) or dormant canes (3 per grapevine) were collected in autumn (March or May) or in winter (June–August), respectively. When collecting samples, canes or petioles were randomly selected from across the whole canopy of each grapevine. Samples were stored at 4 °C until they were processed.

### 2.3. Nucleic Acid Extraction

#### 2.3.1. Nucleic Acid Extraction for RT-PCR

Total nucleic acid (TNA) extraction for virus detection by RT-PCR was performed as follows. A stock silica slurry was prepared by washing one gram of silica (Sigma-Aldrich, cat.S5631, Darmstadt, Germany) with PCR-grade water (Thermo Fisher Scientific, Waltham, MA, USA), centrifuging at 6900× *g*, discarding the supernatant and resuspending the pellet in 1.5 mL of guanidine hydrochloride (GHC) extraction buffer (4 M guanidine hydrochloride, 0.2 M sodium acetate, pH 5.0, 0.2 M EDTA, 0.5% (*w*/*w*) PVP-40 (Sigma-Aldrich, Darmstadt, Germany) and 0.5% (*w*/*w*) sodium metabisulfite (Sigma-Aldrich, Darmstadt, Germany). Plant tissue from an equivalent proportion of each sampled leaf or cane were pooled and homogenized in GHC extraction buffer at a ratio of 100 mg tissue to 1 mL buffer. One twentieth volume of 20% (*wt*/*vol*) sarkosyl (Sigma-Aldrich, Darmstadt, Germany) was added to the homogenate and incubated at 65 °C for 10 min. The homogenate was mixed with one-third volume of chloroform: isoamyl alcohol mixture (24:1, Sigma-Aldrich, Darmstadt, Germany) and centrifuged at 6900× *g* for 10 min. Then, 350 μL of 100% ethanol was added to 500 μL of clear supernatant and mixed with 30 μL of the GHC silica slurry. The sample was mixed for 5 min on a rotating wheel to allow the TNA to bind to the silica particles. Silica was then pelleted and washed once in 700 μL of GHC buffer without PVP40 and twice with 1 mL of wash buffer (75% ethanol, 10 mM Tris-HCl pH 7.5, 100 mM LiCl). Pellet was dried at 65 °C for 1 min then left at ambient temperature until completely dried. This was then resuspended in 50 μL of elution buffer (10 mM Tris-HCl, pH 8.5) and incubated at 65 °C for 1 min. Silica was then pelleted by centrifugation and supernatant transferred to a fresh tube and stored at −20 °C.

#### 2.3.2. Nucleic Acid Extraction for HTS

For HTS, ribosomal RNA depleted double-stranded RNA (dsRNA) was used. The dsRNA was extracted from each sample using a previously published CF-11-based method [22] that was modified. Briefly, 3 g of tissue was homogenized in 8 mL of GHC buffer. About 7.5 mL of homogenate was mixed with one-third volume of chloroform: isoamyl alcohol mixture (24:1, Sigma-Aldrich, Darmstadt, Germany) and centrifuged for 10 min at 4200× *g*. Seven mL of the clear homogenate was transferred into a 15 mL centrifuge tube, and the volume adjusted to 20% ethanol (ethanol: homogenate ratio 1:4 *vol*/*vol*). Then 0.1 g of Whatman^®^ CF-11 cellulose powder (Sigma-Aldrich, Darmstadt, Germany) was added to the homogenate and the mixture was rotated for 10 min. The mixture was centrifuged at 4200× *g* for 2 min and supernatant discarded. The CF-11 was washed and centrifuged twice with 5 mL of STE/20% ethanol buffer (20% ethanol (*vol*/*vol*), in 10 mM Tris-HCl pH 8.0, 100 mM NaCl and 1 mM EDTA). A final 3 mL of 80% STE buffer (in 20% ethanol) was added to CF-11 and vortexed to resuspend. The CF-slurry was dispensed into a syringe containing 0.2 gm of autoclaved glass wool fibre (SiO_2_) and the CF-11 was dried by gently pressing the plunger to remove any remaining buffer. The dsRNA was eluted in 1 mL of preheated (65 °C) 1× STE buffer.

#### 2.3.3. Nucleic Acid Quality Control

Prior to RT-PCR and Meta-HTS, the quality, integrity and quantity of RNA for RT-PCR and dsRNA for HTS were evaluated using a Nanodrop TM 1000 spectrophotometer (Thermo Fisher Scientific, Waltham, MA, USA) and a Qubit fluorometer (Thermo Fisher Scientific, Waltham, MA, USA), according to the manufacturer’s instructions. Samples with the OD 260/280 and OD 260/230 values outside of the acceptable range were re-extracted (260/280 >1.8, 260/230 ≈ 2.0). In addition, all RNA used for virus detection by RT-PCR was tested using the RubiscoL internal control RT-PCR assay (Appendix A) [23] to ensure RNA was present and detection was not affected by inhibitors. Each Meta-HTS sample was tested for expected viruses including GVA, GLRaV-1, GLRaV-3, GLRaV-4 strain 6 and 9, grapevine rupestris stem pitting-associated virus (GRSPaV) using virus specific RT-PCR assays listed in Appendix A.

### 2.4. RT-PCR Reaction Conditions

A total of 10 µL RT-PCR reactions contained 4 units of ProtoScript^®^ II reverse transcriptase (New England Biolabs, Ipswich, MA, USA), 0.15 units of EpiMark^®^ Hot Start Taq DNA polymerase (New England Biolabs, Ipswich, MA, USA), 2 µL 5XGoTaq^®^ Flexi Reaction Buffer (Promega, Madison, WI, USA), 0.2 mM of each dNTP, 1.5 mM of MgCl_2_, 0.2 µM of each forward and reverse primer, PCR-grade water (Thermo Fisher Scientific, Waltham, MA, USA) and 1 µL of TNA. Reaction conditions were as follows: reverse transcription at 43 °C for 45 min and initial denaturation at 95 °C for 30 s, followed by 35 cycles of denaturation at 95 °C for 20 s, annealing (assay dependent, see Appendix A) for 20 s, and extension at 68 °C for 30 s, and then a final extension at 68 °C for 5 min.

### 2.5. Metagenomic High-Throughput Sequencing (Meta-HTS)

#### 2.5.1. Nucleic Acid Pretreatments, Library Preparation and Sequencing

Prior to library preparation, dsRNA was concentrated using isopropanol and sodium acetate as previously described [24]. The concentrated dsRNA was DNase-treated to remove DNA using the Ambion™ DNase I kit (Invitrogen, Waltham, MA, USA) according to manufacturers’ instructions. 

Libraries of the Meta-HTS samples were prepared using the TruSeq Stranded Total RNA with Ribo-Zero Plant (Illumina, San Diego, CA, USA) kits, following the manufacturer’s instructions. The concentration of libraries was determined using a Nanodrop, Qubit fluorometer, and TapeStation (Agilent, Santa Clara, CA, USA) and pooled to equivalent concentration. The pooled library was sequenced using a NovaSeq instrument with 2 × 150 bp read length. 

#### 2.5.2. *De Novo* Assembly

Illumina adapters were trimmed and the raw reads with a quality score below 20 and length below 50 bp were removed using TrimGalore (v. 0.4.2) [25]. Trimmed reads were *de novo* assembled using SPAdes (v. 3.12.0) with default settings [26]. Assembled contigs were compared against the local database built with the exemplar isolate of each virus species identified by the International Committee on Taxonomy of Viruses (ICTV) and latest release of the viral sequences from the GenBank database using the “makeblastdb” function of BLAST+ (v. 2.11.0). The contigs were blasted against the local database to obtain a list of virus species in each sample using the “blastn” function [27]. The number of raw and quality trimmed reads, number of contigs and contig length were obtained using the “stats” and “readlength” functions of the BBMap (v. 35.85). To obtain the average coverage of each *de novo* assembled contig (complete genome sequence used in the phylogenetic analysis) of each virus, quality trimmed reads were mapped back to each *de novo* assembled virus contig using the “bbmap” function of the BBMap [28]. 

#### 2.5.3. Phylogenetic and Sequence Similarity Analysis

GVA, GLRaV-3 and GLRaV-4 contigs longer than 7000 nucleotides (nts), 17,000 nts and 13,000 nts, respectively, were used for phylogenetic analysis of each virus. Each virus contig was aligned to all publicly available GenBank complete genome sequences (Appendix A) using Muscle (v.3.8.31) [29]. Nucleotide (nt) identity and amino acid (aa) similarity were determined using the sequence demarcation tool (SDT, v. Linux64) [30]. Phylogenetic trees were constructed using the neighbour-joining method with 1000 bootstrap replicates by MEGA (v. 7.0.26) [31]. The neighbour-joining method was used to enable comparison of GVA phylogenetic groupings reported in previous studies [11]. Phylogenetic trees of GVA were constructed using complete genomes as well as nt and aa sequences from all gene regions excluding ORF2. The phylogenetic groups of GVA contigs were assigned based on the clades of the phylogenetic tree of the full genomes and the coat protein (CP) gene. Phylogenetic trees of GLRaV-3 were constructed using complete genome sequences and complete nt sequences of the CP gene. Phylogenetic trees of GLRaV-4 were constructed using complete genome sequences and complete aa sequences of the RdRp, heat shock protein 70 homologue (HSP70h) and CP genes.

#### 2.5.4. Phylogenetic Group Identification of the Grapevine Virus A Contigs 

When phylogenetic groups of the complete genome sequences from both GenBank and Meta-HTS experiments were determined using the phylogenetic analysis described above, they were used as a reference database by BLAST+ to build a local blast. All short contigs were blasted against the reference database using the “blastn” function [27]. The phylogroup of each short contig was obtained by the best match that gave the highest percentage nucleotide identity to this contig. 

#### 2.5.5. Multiple Sequence Alignment of the GVA RNA-Binding Protein 

The GVA RB gene nucleotide sequence was translated to amino acid sequence using the ”translate to protein” function and aligned to coding protein sequences from GenBank using the “create alignment” function within the CLC Genomics Workbench (v. 21.0.3; Qiagen, Aarhus, Denmark). The SD status of each GenBank isolate was obtained from publications [11,19,32,33] and listed in Appendix A.

#### 2.5.6. Recombination Analysis

Complete or near-complete viral contigs of GVA, GLRaV-3 and GLRaV-4 were combined with all publicly available full genome sequences from GenBank (Appendix A) and aligned using Muscle (v. 3.8.31). The sequence alignment was trimmed according to the shortest sequences and analysed by RDP5 (v. Beta 5.23) [34]. Seven methods RDP [35], GENECONV [36], Chimaera [37], MaxChi [38], BootScan [39], SiScan [40], 3Seq [41] were selected to detect recombination events. If more than four out of seven methods detected the same recombination event for a contig, it was considered as a recombined sequence and excluded from the phylogenetic analysis.

## 3. Results

### 3.1. Virome Analysis by Endpoint RT-PCR and Meta-HTS

A summary of the overall number of grapevines with identical virus status by RT-PCR is given in Appendix A. The virus status of each individual grapevine by RT-PCR but not by Meta-HTS can be found in Appendix A. The virus status by both methods can be found in Appendix A. The viruses detected by RT-PCR and/or Meta-HTS in grapevines across WIL, LC, BV and CV were GVA, GLRaV-1, GLRaV-3, GLRaV-4 (including strains 4/5, 4/6 and 4/9), grapevine rupestris vein feathering virus (GRVFV), GRSPaV, grapevine virus F (GVF) and grapevines red globe virus (GRGV).

#### 3.1.1. RT-PCR

Across all Shiraz grapevines from WIL, LC and BV (*n* = 116), the viruses that were detected by endpoint RT-PCR included, GVA (51/116), GLRaV-1 (3/116), GLRaV-3 (50/116), GLRaV-4/6 (34/116), GLRaV-4/9 (52/116), GRVFV (76/116) and GRSPaV (116/116). GRVFV was found in 72/80 (90%) grapevines in the WIL vineyard but only 4/30 (13.3%) at LC. The frequency of GLRaV-4 strains 6 and 9 at WIL was 33/80 (41.25%) and 43/80 (53.75%), respectively (Appendix A). At the LC site, GLRaV-4 strain 6 only occurred in 1/30 (3.3%) grapevines and GLRaV-4 strain 9 was detected in 9/30 (30%) of the total tested grapevines. GLRaV-1 was detected at BV in 3/6 grapevines with mild LRD symptoms, which also had GVA and GRSPaV, but it was not detected at any other site. GLRaV-3, GLRaV-4 strain 6 or 9, GRSPaV and GRVFV may also be present in SD-affected grapevines, but they were not consistently associated with SD, or they were also found in grapevines without SD (Appendix A).

In the 16 SD-affected Shiraz grapevines at the WIL site, 16/16 were positive by the general purpose GVA endpoint RT-PCR assay using primer pairs Ah587/Ac995 and H7038/C7273 that detect phylogroups I and II, and 5/16 were also positive using the phylogenetic group III (GVA^III^) specific endpoint RT-PCR assay. Among 35 LRD and 29 asymptomatic Shiraz grapevines, none were positive by the general purpose GVA assays (Table 2 and Appendix A). A total of 12/35 LRD and 5/29 asymptomatic Shiraz grapevines were positive by the GVA^III^ assay (Table 2 and Appendix A).

For the Cabernet Sauvignon and Merlot grapevines from WIL and CV, 6/6 were positive by the GVA general purpose endpoint RT-PCR assay and 5/6 were positive by the GVA^III^ specific RT-PCR (Table 2 and Appendix A). The two Shiraz grapevines from CV were positive for GRSPaV and GRVFV only.

Grapevine yellow speckle viroid 1 and hop stunt viroid were frequently detected in most of the samples by Meta-HTS (Appendix A), but no further analysis was performed on the viroids.

#### 3.1.2. Comparison of Virome Detection between Meta-HTS and Endpoint RT-PCR

The viromes from a total of 50 grapevines, including 43 Shiraz grapevines across all vineyards, including WIL (*n* = 24), LC (*n* = 14), BV (*n* = 3) and CV (*n* = 2), plus one Merlot grapevine from WIL and six Cabernet Sauvignon grapevines from WIL (*n* = 4) and CV (*n* = 2), were analysed by Meta-HTS and compared with the endpoint RT-PCR results (Table 2). For the presence of GVA strains, GLRaV-3 and the GLRaV-4 strains, comparable results were obtained in 40/50 grapevines. Meta-HTS detected GVA and GLRaV-4 in 6/50 grapevines that were missed by RT-PCR, and the presence of GLRaV-3 was missed by Meta-HTS in 3/50 grapevines (Table 2).

Near-complete or partial genomes of GVA^II^ were obtained from 21/50 and 7/50 grapevines, respectively, including all SD-affected shiraz grapevines (*n* = 29) at WIL and LC, one SD-affected Merlot grapevine at WIL and all six LRD-affected Cabernet Sauvignon SA125 grapevines from WIL (4/6) and CV (2/6) (Table 2). In 10/50 grapevines, two distinct partial or near full-length genomes of GVA were assembled (Appendix A).

The detection of GVA by Meta-HTS generally corresponded with the results of the endpoint RT-PCR assays with some exceptions. This included the designation of phylogroup type based on sequence comparison of whole genomes and comparisons of the RdRp, MP and CP genes (Table 2, Figure 2). Meta-HTS showed that GVA^II^ isolates were always associated with SD in Shiraz and Merlot, except in WIL 47 (Shiraz) where GVA^III^ was detected by Meta-HTS, but the RT-PCR results suggested phylogenetic group I (GVA^I^) or GVA^II^ were present (Table 2). GVA was also detected in nine LRD-affected Shiraz grapevines from WIL (7/9) and BV (2/9) by Meta-HTS and RT-PCR (Table 2). Seven out of nine were positive by Meta-HTS and 6/7 of the same grapevines by RT-PCR (Table 2). Meta-HTS indicated GVA^I^ was present in the two mild LRD-affected grapevines at BV and GVA^III^ was present in the five LRD-affected grapevines at WIL (Table 2). GVA was not detected in any of the asymptomatic Shiraz grapevines from WIL, LC, BV and CV (Table 2). GVA^II^ and GVA^III^ were detected in 6/6 and 5/6 LRD-affected Cabernet Sauvignon grapevines (Table 2).

GLRaV-3 was detected in all SD- affected grapevines (*n* = 13) at WIL by RT-PCR, but genomes were only assembled from 10/13 of the affected grapevines. GLRaV-3 was also detected from the single SD-affected Merlot and all LRD-affected Cabernet Sauvignon (*n* = 4) grapevines at WIL by both methods. GLRaV-3 was not detected in SD-affected or asymptomatic grapevines from LC and Cabernet Sauvignon grapevines from CV. Genomes and RT-PCR confirmed the detection of GLRaV-3 in the same LRD-affected grapevines at WIL and the detection of GLRaV-1 in mild LRD-affected grapevines at BV.

At WIL, GLRaV-4 strains were detected by RT-PCR and genomes assembled in the same 8/13 SD-affected Shiraz grapevines and in all seven LRD-affected Shiraz grapevines. Additionally, GLRaV-4 strains were detected by RT-PCR and genomes were assembled in 2/4 and 3/4 asymptomatic Shiraz grapevines. GLRaV-4 was detected by RT-PCR and genomes assembled in 5/9 and 9/9 SD-affected grapevines at LC, but it was not detected in asymptomatic grapevines. GLRaV-4 was also found in all LRD-affected Cabernet Sauvignon grapevines at WIL and CV by RT-PCR and Meta-HTS.

GRSPaV was detected in all 50 grapevines by RT-PCR and Meta-HTS (Appendix A and Appendix A). GRVFV was detected by RT-PCR and by Meta-HTS in 24/50 and 32/50 grapevines, respectively, including SD, LRD and asymptomatic grapevines from WIL, LC, BV and CV (Appendix A). GRGV was not tested using RT-PCR but was found by Meta-HTS in five LRD-affected Cabernet Sauvignon grapevines from WIL and CV (Appendix A). GVF was only found from the WIL site in 2/4 of the Cabernet Sauvignon grapevines, but not in the Cabernet Sauvignon from CV by Meta-HTS only (Appendix A).

Two own-rooted asymptomatic Shiraz BVRC12 grapevines at CV did not have the same virus profile as grapevines from WIL and LC and only GRSPaV and GRVFV were detected by Meta-HTS (Appendix A).

### 3.2. Phylogenetic Analysis of Grapevine Virus A (GVA)

#### 3.2.1. Phylogenetic Groups (Phylogroups) of Grapevine Virus A (GVA)

The GVA isolates used for analysis are listed in Appendix A. Three major GVA phylogroups, GVA^I^, GVA^II^ and GVA^III^, were formed when phylogenetic analysis was used to compare the nucleotide sequence of the 35 whole genomes and the RdRp, CP and MP genes from the same 35 Australian GVA isolates and isolates available in GenBank (Figure 2a,b,d,f; Appendix A). Three clusters were also formed when amino acid sequences of the RdRp, CP and MP gene products were analysed (Figure 2c,e and Appendix A). In most cases, the phylogenetic clade designation of each Australian isolate was the same when comparing across all gene nt or aa sequences (Figure 2 and Appendix A; Appendix A). However, some isolates designated as group I based on the nt sequence of the CP gene (Figure 2d) shifted into the group II clade when the corresponding aa sequences were compared (Figure 2e).

Only two distinct clusters were formed when phylogenetic analyses were used to compare the nt (Figure 2g) and aa sequences (Appendix A) of the RB gene. The first cluster comprised of the isolates that were previously identified as GVA^I^ and GVA^II^ using the CP gene, and the second RB cluster included those previously identified as GVA^III^ using the CP gene.

#### 3.2.2. Similarity within Phylogroups and between Australian GVA Isolates

The range of percentage nt identities and aa similarities within each GVA phylogroup for whole genomes, RdRp, CP, MP and RB genes within Australian isolates from this study, and all previously identified Australian and international isolates, is given in Table 3. The data show that all isolates within GVA^I^, GVA^II^ and GVA^III^, at the full genome level, share 76.33–99.8%, 79.92–99.90% and 76.52–99.94% nt identities, respectively. Isolates within Australia share nt identities between 91.45–99.94% at full genome level. Among all Australian isolates, the lowest nt identities and aa similarities are 90.65% and 91.45%, respectively, when RdRp, CP, MP and RB genes were analysed. The sequence similarity between all available isolates across all three phylogroups of RdRp, CP, MP and RB genes were 74.76–99.96% nt and 84.72–100% aa (RdRp), 77.30–100% nt and 78.85–100% aa (MP), 77.39–100% nt and 79.90–100% aa (CP), and 86.08 to 100% nt and 84.62–100% aa, identities and similarities, respectively. Only two Australian isolates belonging to the GVA^I^ phylogroup were found, BV1_N31_mild_LRD and BV1-1 (accession no. MT070961), both from the BV vineyard (Table 3 and Appendix A) [1]. The highest nucleotide identity shared between local and international GVA^I^ isolates was 90.76% and this was between Australian isolate BV1_N31_mild_LRD and French isolate TT2017-79 (accession no. MK404722) from Pinot Noir (Appendix A).

### 3.3. Association between Amino acid Sequence of RNA-Binding Gene and Symptom Expression of Grapevine Virus A

Amino acid sequences of the RB gene of GVA isolates obtained from Meta-HTS in this study, and GVA isolates associated with SD from previous studies [11,19,32,33] were aligned and the results are shown in Figure 3. All Australian GVA^III^ isolates from this study have an extra four aa residues at the c-terminus, whereas the two GVA^III^ isolates from South Africa, P163-1 and GTR1-1, and other GVA^I^ and GVA^II^ isolates, lack these residues. No association between symptom expression and amino acid residue changes were observed consistently between GVA^I^ and GVA^II^ isolates from SD-affected and -unaffected grapevines. GVA^III^ isolates consistently had a leucine residue at position 31, which was not found in GVA ^I^ and GVA^II^ isolates (Figure 3), and a glutamic acid residue at position 61 that was only found in one GVA^II^ isolate (WIL2_N36_SD).

### 3.4. Phylogenetic Analysis of Grapevine Leafroll-Associated Virus 3

The nucleotide sequences of 77 near-complete genome sequences from GenBank and 21 genome sequences from this study were analysed (Figure 4 and Appendix A). The phylogenetic trees obtained from the complete genome sequence (above 17,027 nts) and CP gene alignments both show five major phylogenetic groups. Group names were redefined based on the genetic distance to the exemplar isolate NY1 (nt identities high to low) of the full genome sequences (Figure 4a) and CP genes (Figure 4b). The 21 Meta-HTS sequences of this study share 99.86 to 99.99% nt identities with each other using alignment of the complete genome sequences of 18,1713 nts. They all clustered into group I with the exemplar isolate NY1 from USA. Phylogenetic analysis of the RdRp and HSP70h genes confirmed the five-group system. The CP sequences within phylogroup I, II, III, IV and V share 94.59–99.47%, 92.46–92.78%, 91.4–91.72%, 81.1–82.70% and 74.31–79.83% nt identities, respectively, when compared to the isolate NY1 (Appendix A). The standard length of the GLRaV-3 CP gene is 942 nts, with only one unique isolate, 3m-139 (accession no. JX266782) from *Vitis vinifera* cv. Sauvignon Blanc from Australia has an extra 15 nts insertion compared to other isolates. This isolate is believed to be an asymptomatic variant of GLRaV-3 [42,43].

### 3.5. Phylogenetic Analysis of Grapevine Leafroll-Associated Virus 4, Strains 5, 6 and 9

The Meta-HTS assembled a total of 35 near-complete genome sequences of GLRaV-4 isolates from 31 grapevines including 5/31 that had two distinct strains, and they were compared with all complete genome sequences available in the GenBank database. The phylogenetic analysis of the complete genome sequence and aa sequences of the RdRp, HPS70h and CP gene showed five major clades, which represent strains 4, 5, 6, 9 and 10 (Figure 5). Appendix A shows the amino acid pairwise similarities of the RdRp, HPS70h and CP genes for each isolate with the exemplar isolate LR106 (accession no. FJ467503). GLRaV-4 strain 9 is more frequently found among all Meta-HTS sequences of GLRaV-4 (26 out of 36), followed by strain 6 (8 out of 36) (Figure 5; Appendix A). GLRaV-4 strain 5 is the most uncommon strain that was only found in three grapevines at WIL (WIL13 to WIL15) (Appendix A). The two unique strains Ob and Car are phylogenetically distantly related to other strains used for analysis, as indicated by the star symbols in Figure 5a. Strain Ob shares the lowest aa similarity of 72.54% with the exemplar isolate LR106 in the RdRp gene (Appendix A). It is nine amino acids longer than other GLRaV-4 strains (527 aa compares to 518 aa). The isolate Car has one amino acid extra in the HSP70h protein compared to other isolates (535 aa compares to 534 aa). By pairwise similarity matrix, Ob and Car had the lowest pairwise aa similarity to other isolates in the HSP70h gene, ranging from 66.17 to 71.35% (Appendix A).

Based on the length of the CP aa sequence, all isolates of strain 4 and strain 10 had 273 aa, all isolates of strains 5 and 9 had 268 aa and all isolates of 6 had 269 aa. The CP gene protein homology shows that strains 5, 6 and 9 are phylogenetically more similar than strains 4, 10, Car and Ob. Within strain 4, 5, 6, 9 and 10, pairwise aa similarity of the CP gene to the exemplar isolate LR106 (accession no. FJ467503) shares 99.26–99.63%, 81.02–83.21%, 79.93–80.66%, 81.39–82.48% and 77.01% (only one isolate), respectively, using similarity matrix of all GenBank available isolates and Australian isolates listed in Appendix A.

### 3.6. Recombination Analysis

Recombination analysis was performed for all Meta-HTS sequences along with complete genome sequences from GenBank (Appendix A) used in phylogenetic analysis. No recombination events were detected in GVA, GLRaV-3 and GLRaV-4 sequences obtained from this study.

## 4. Discussion

### 4.1. Association between SD and Phylogenetic Groups of GVA

In this comprehensive virome study, further evidence of the association between GVA^II^ variants and SD in Australia was shown, supporting previous studies both in Australia and South Africa [10]. This study indicated that GVA^II^ variants might also be associated with SD in Merlot, although only one grapevine was analysed. Although GVA^III^ variants were often found in mixed infections with GVA^II^ variants, the results presented in this study suggest they are unlikely to be associated with SD symptoms on Shiraz because they were also present in LRD and asymptomatic grapevines in the absence of GVA^II^ variants (Table 2 and Appendix A). GVA^III^ was only found in one SD-affected grapevine (WIL47) with Meta-HTS, but this sample was positive by endpoint RT-PCR, which detects GVA^I^ and GVA^II^ variants. Therefore, it is possible that a GVA^II^ variant was present in WIL47 but the titre was too low for detection by Meta-HTS. GVA^I^ was only found in a few LRD-affected grapevines from BV, and in this study, was not associated with SD.

Although a strong association between GVA^II^ and SD was observed, it is possible that some GVA^I^ and GVA^III^ variants could cause SD, and that some GVA^II^ variants may not. For example, isolate GTR1-2 from South Africa was identified as SD-negative and induced mild vein clearing symptoms on *N. benthamiana* [11] and yet it belongs to group II (Figure 2a). This is likely due to the ORF1, ORF3, 5′NTR and 3′NTR, which were clearly divergent from other GVA^II^ isolates [11]. Although the current data suggest that GVA^II^ variants are most likely to be the key pathogen of SD in Australia, phylogenetic grouping of a variant should not be used alone to predict virulence. Broader surveillance of SD-sensitive varieties, with and without SD, across multiple grape-growing regions should be performed to further confirm the association between specific GVA phylogroups and SD.

### 4.2. Amino Acid Sequence of RNA-Binding Gene of GVA and Symptom Expression

ORF5 of GVA, which encodes a 10 kDa RNA-binding protein, was reported to play an important role in symptom expression on *N. benthamiana*. When ORF5 of the infectious clone of isolate PA3 was modified, this virulent clone no longer induced symptoms on *N. benthamiana* [20]. Later, it was demonstrated that an association between symptom severity in *N. benthamiana* plants inoculated with seven South African GVA isolates was dependent on the eighth amino acid sequence from the N-terminus [22]. Therefore, we investigated if this change could be involved in SD symptom expression in Australian isolates, but it was not demonstrated in this study. Goszczynski and Habili [10] reported the South African GVA^II^ isolate, P163-M5 (accession no. DQ855082), has a 119 nt insertion on ORF 2 and this isolate induced stronger leaf mottling symptoms on *N. benthamiana* compared to other South African GVA isolates. ORF 2 encodes a putative 19-kDa protein with no significant sequence identity to other proteins in the database. Only one other GVA^II^ isolate, M5v (accession no. MK982553), has this insertion at the same position. This study therefore illustrates that this insertion is not associated with SD symptom expression in Australian isolates. This study did highlight residues at positions 31 and 61, which differentiated the GVA^III^ phylogroup from GVA^II^ isolates consistently, and GVA^I^ isolates in most cases (Figure 3). This may highlight important residues that could impact the virulence of GVA; however, further studies are required to test this hypothesis. Further comparative genomic analyses and functional genomic studies are required to investigate the specific sequences that interact with the host to cause SD in sensitive varieties.

### 4.3. GVA Diversity

Phylogenetic relationships between GVA^I^ and GVA^II^ isolates, based on genomes and individually analysed RdRp, MP and CP genes, indicate that they are more closely related to each other than either group are to GVA^III^. They share the same primary branch with ≥99% bootstrap support (Figure 2 and Appendix A, in red square), and cannot be differentiated using the neighbour-joining trees of the RNA-binding gene by both nt and aa sequence (Figure 2g and Appendix A). In addition, some GVA isolates were placed within the GVA^I^ clade when analysing nt sequences of the CP gene but placed within the GVA^II^ clade when using the aa sequence of the same gene (Figure 2e). This all supports the hypothesis that GVA^I^ and GVA^II^ phylogroups may have diverged from the same ancestor.

Diversity between GVA^II^ and GVA^III^ strains was observed in Shiraz, Cabernet Sauvignon and Merlot grapevines at WIL, although some Shiraz grapevines had GVA strains that were almost identical (Figure 2a). When the longest GVA^III^ contigs from each of the Cabernet Sauvignon, Merlot and Shiraz samples were analysed by pairwise nt identity, the identities of GVA^II^ and GVA^III^ contigs were between 91.05–99.96% and 95.04–99.95%, respectively (Appendix A). This suggests GVA infection may have originated from multiple sources. One Cabernet Sauvignon grapevine (Cabw12) had two distinct GVA^II^ strains, and both strains clustered with GVA^II^ strains from SD-affected grapevines at WIL and LC but were not identical (Figure 2a and Appendix A). Therefore, although some spread may have occurred between Shiraz grapevines and adjacent vineyards, infections are also likely to have been introduced from other sources.

### 4.4. GLRaV-3

As a part of this study, the diversity of GLRaV-3 and GLRaV-4 were also investigated in SD and asymptomatic grapevines due to previous observations that these viruses may play a role in this disease [1]. Using all available GLRaV-3 sequences from GenBank, and Australian isolates from this study, a tree with five phylogroups was produced and new naming of these proposed based on the aa sequence similarity of the CP gene from high (phylogroup I) to low (phylogroup V) to the exemplar isolate NY1. This finding contrasts to several previous studies, including our preliminary study, which identified seven phylogroups [1], and that described by Diaz-Lara et al. (2018) that identified 10 phylogroups [44]. It appears the addition of a larger number of full-length GLRaV-3 genomes and CP gene sequences in our most recent phylogenetic analysis has more clearly defined the evolutionary relationship between isolates, resulting in the collapse of previously described phylogroups to a smaller number. This proposed grouping was also reflected in one previous study, which used all available GenBank sequences of the full-length CP gene when assessing GLRaV-3 isolates obtained from Portuguese grapevine varieties [45]. This study demonstrated that the phylogroup of any novel GLRaV-3 isolates could be identified using any of CP, RdRp, HSP70h or full-length sequences since they provided consistent results (Figure 4 and Appendix A).

In this study, low diversity between all Australian GLRaV-3 strains was observed that were all isolated from grapevines located in the WIL vineyard. These all clustered in phylogroup I, including those infecting Shiraz, Cabernet Sauvignon SA125 (Cabw1, 2, 11, 12) and Merlot (Merlot 1), and shared 99.86 to 99.99% nt identity. This suggests a single origin of the virus at the WIL site, potentially from the same source, and then transmitted by natural spread through insect vectors. According to the phylogenetic analysis of GLRaV-3, all Australian GLRaV-3 isolates clustered with the NY1, which could indicate they might have originated from the USA.

### 4.5. GLRaV-4

GLRaV-4 species are the most genetically diverse groups within the *Ampelovirus* genus, and GLRaV-4 strains 4/5, 4/6, 4/9, 4/10, 4/Car, 4/De and 4/Pr were previously thought to be independent virus species. In 2012 they were classified by Martelli et al. [4] into a single species based on their serological relationships, genome structure, size, and biological and epidemiological characteristics. Strains Ob and Car have been classified as strains of GLRaV-4, but they have lower homology to other strains of this virus [46,47].

In this study, genomes of Australian isolates of GLRaV-4/5, -4/6 and -4/9 are reported for the first time. GLRaV-4 strains 6 and 9 in Shiraz from WIL and LC showed low diversity and close phylogenetic relationships to the GLRaV-4 isolates in Cabernet Sauvignon from WIL (Figure 5). This suggests the GLRaV-4 in Shiraz from WIL and LC may have originated from the same source, possibly the Cabernet Sauvignon clone SA125.

### 4.6. Other Viruses

Other viruses that were detected in grapevines in this study included GLRaV-1, GRVFV, GRSPaV, GVF and GRGV. GRVFV and GRSPaV were two abundant viruses in all the South Australian vineyards studied and were found in SD- and LRD-affected and -unaffected grapevines. They are therefore, not considered to be associated with SD or LRD and were not studied further. GLRaV-1, GVF and GRGV were infrequently found and also not considered to be associated with SD. This is the first report of GVF and GRGV in Australia. The detection of these two viruses will be described in a different paper in the near future.

### 4.7. Variability in Virus Detection

Variability in the detection of some viruses by Meta-HTS and RT-PCR was observed. The detection of viruses by Meta-HTS and not by RT-PCR is most likely due to sequence variability at the RT-PCR primer binding sites. For example, in this study, the primer pair H7038/C7273 from Goszczynski and Jooste [33], that were claimed to detect all GVA strains, only detected strains GVA^I^ and GVA^II^, and a third assay was required to detect GVA^III^ variants. The failure to detect GLRaV-4/9 and GRVFV in some samples by Meta-HTS could be due to low virus titre in the extracts, which could be below the detection threshold due to seasonal fluctuation or uneven distribution of the virus in the sampled tissue [48,49].

### 4.8. Association between SD and LRD Species

In all SD-affected grapevines, at least one grapevine leafroll virus species, either GLRaV-3 or GLRaV-4, was also present with GVA (Table 2, Appendix A and Appendix A). This may be associated with the enhanced transmission efficiency of GVA by insect vectors in the presence of GLRaVs [50]. It is possible that the presence of a GLRaV species is important in SD symptom expression; however, there was no association to a specific GLRaV species. Additionally, strains of GLRaV-3 and GLRaV-4 were found in LRD-only-affected grapevines and asymptomatic grapevines, suggesting they are not the sole cause of SD.

In conclusion, this study showed a strong association between GVA^II^ variants and SD. SD has only been reported from two countries, Australia and South Africa. This distribution could be due to the low prevalence of SD-related GVA^II^ variants in the USA and other countries (Appendix A). Another reason for the prevalence of SD in Australia could also be due to more conducive environmental conditions or more efficient insect vectors. In this study, clone SA125 of Cabernet Sauvignon was tolerant to infection by GVA^II^ as SD symptoms were not observed. This clone is one of the most widely planted Cabernet Sauvignon clones in South Australia and could present a risk for the spread of GVA and SD disease but could also pose a risk if infected grapevines are top worked with sensitive varieties, as has previously been observed [51]. Further work to investigate the prevalence and relationships between strains within and between vineyards to estimate the risk of the disease in other regions is still required. A better understanding of vector efficiency for the transmission of GVA and GLRaVs is also required. This highlights the importance of pathogen testing and the provision of high-quality planting material to ensure disease-free sustainable vineyards.

## Figures and Tables

**Figure 1 viruses-15-00774-f001:**
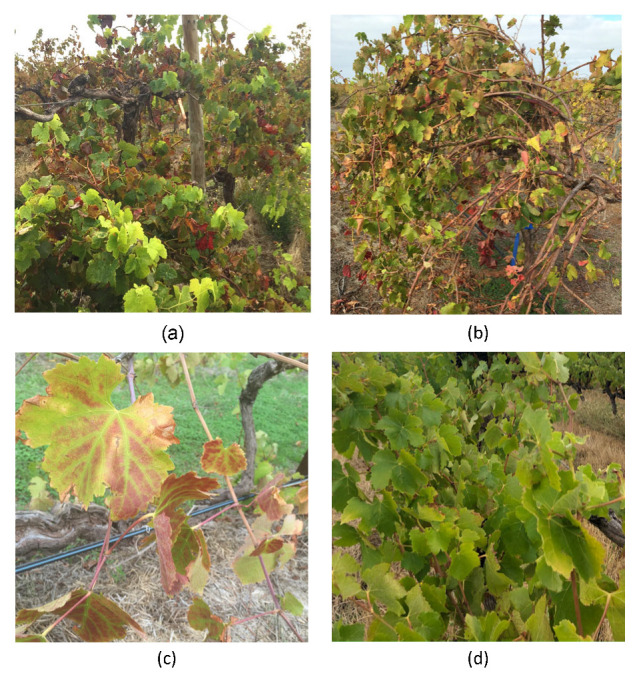
Symptoms of grapevine virus A (GVA)-infected Shiraz from the Willunga, Langhorne Creek and Barossa Valley sites. Symptoms of Shiraz-disease-affected Shiraz grapevines at (**a**) Willunga and (**b**) Langhorne Creek, and infected with GVA^II^ variants (**c**) mild grapevine leafroll disease symptoms on an unknown clone of Shiraz that tested positive to GVA^I^ variant and grapevine leafroll-associated virus 1 at Barossa Valley. (**d**) An asymptomatic Shiraz, clone BVRC12 at Willunga.

**Figure 2 viruses-15-00774-f002:**
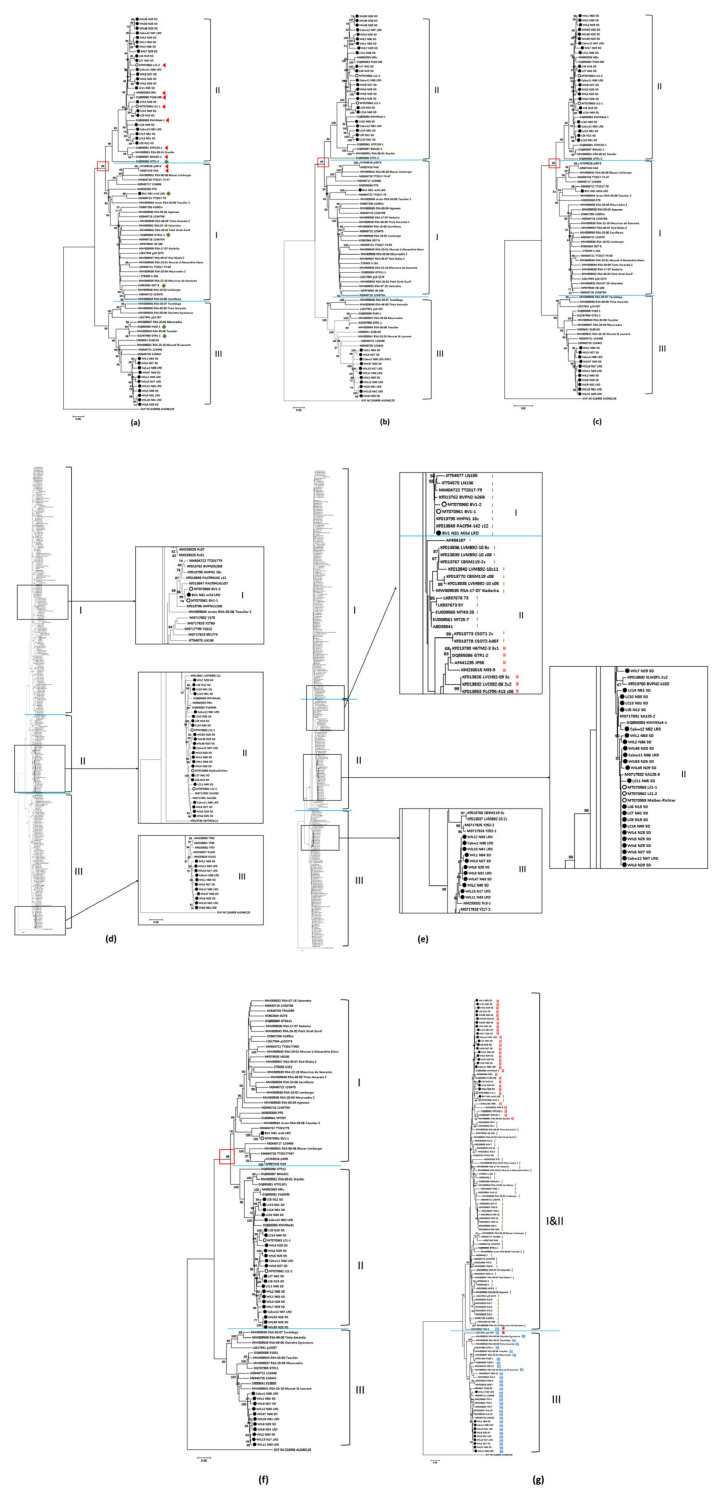
Neighbour-joining phylogenetic analysis of grapevine virus A (GVA) using 35 metagenomic sequences and all available sequences from GenBank. A neighbour-joining tree using (**a**) complete genome sequences alignment above 6991 nts, (GVA isolates associated with SD-affected grapevines are labelled with red triangles and isolates associated with asymptomatic grapevines are labelled with green rhombus), (**b**) full-length nucleotide (nt) sequences of RNA-dependent RNA polymerase (RdRp), (**c**) full-length amino acid (aa) sequences of RdRp, (**d**) nt sequences of complete coat protein (CP) gene, (**e**) aa sequence of CP gene, (**f**) nt sequence of complete movement protein (MP) gene, (**g**) nt sequences of RNA-binding (RB) gene. All phylogenetic trees were constructed using MEGA (7.0.26) software and the neighbour-joining method with 1000 bootstrap replicates. Bootstrap values below 50% were not shown. Australian sequences generated by a previous study by Meta-HTS [1] are labelled using open circles and sequences generated by NovaSeq are labelled by black dots. Isolates for which the phylogroup assigned differed depending on the gene used is marked by * in figure (**g**). The red squares indicate the primary branches which show ≥ 99% bootstrap values. The colour-coded I, II and III labelled in figure (**g**) is based on the phylogroup assigned using the CP gene, for comparison. Grapevine virus F (GVF) isolate AUD46129 (accession no. NC018458) was used as an outgroup.

**Figure 3 viruses-15-00774-f003:**
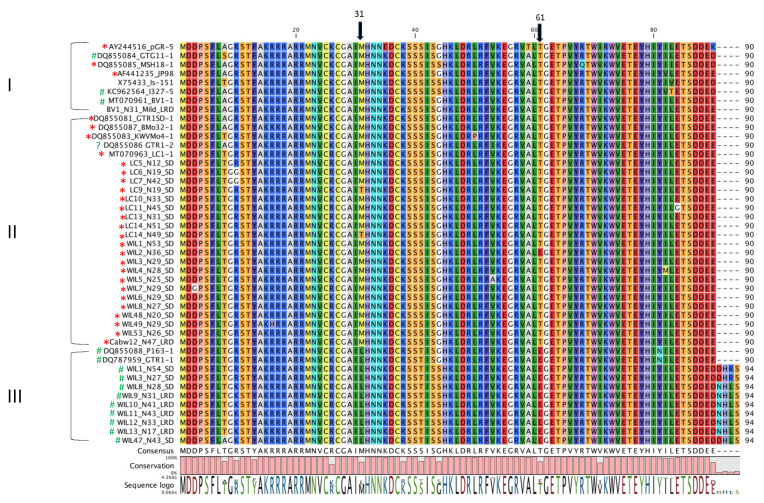
Amino acid alignment of the N-terminal (90–94 aa) of the RNA-binding gene of grapevine virus A (GVA) of Australian and international isolates. The phylogenetic group (I, II or III) of each isolate is on the left. Red * indicates isolates associated with Shiraz disease (SD) and green # indicates isolates not associated with SD. The blue question mark indicates a SD-negative isolate in phylogenetic group II [11]. The two black arrows point to positions 31 and 61 of the alignment.

**Figure 4 viruses-15-00774-f004:**
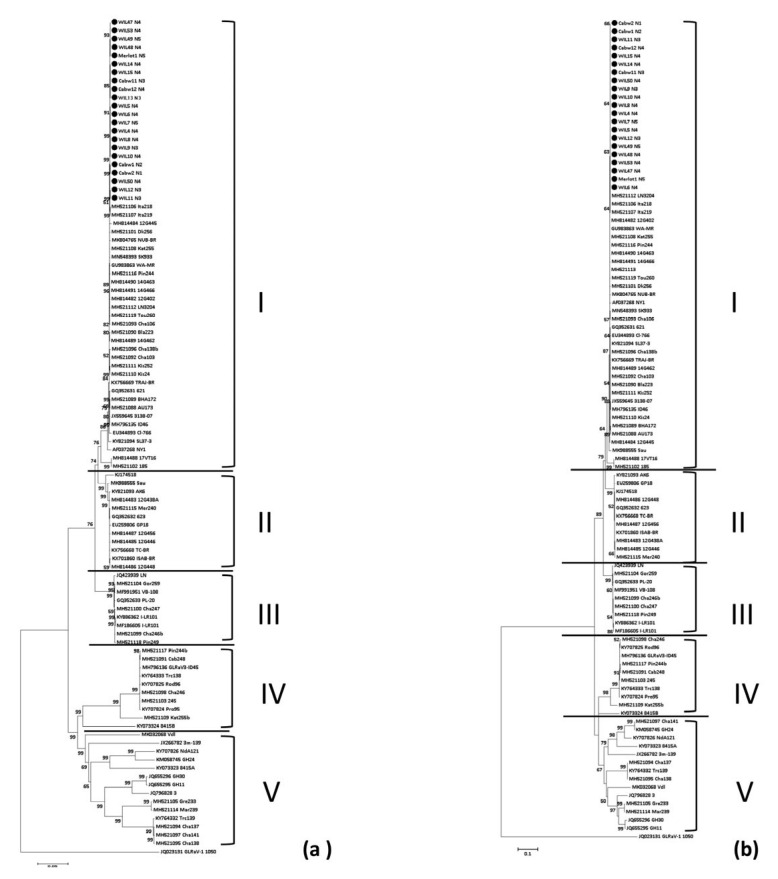
Phylogenetic analysis of 21 near-complete genome sequences (above 17,027 nts) of Australian grapevine leafroll-associated virus 3 (GLRaV-3) isolates and 77 publicly available full genome GLRaV-3 sequences from GenBank. A neighbour-joining tree of (**a**) complete genome sequences, (**b**) nucleotide sequence of the coat protein gene, was constructed with 1000 bootstrap replicates by MEGA software. Black dots denote sequences generated by Meta-HTS of this study. The bootstrap values below 50% are not shown. Sequences generated by this study are labelled by black dots. Roman numerals I–V represent the five distinct GLRaV-3 phylogroups observed in this study.

**Figure 5 viruses-15-00774-f005:**
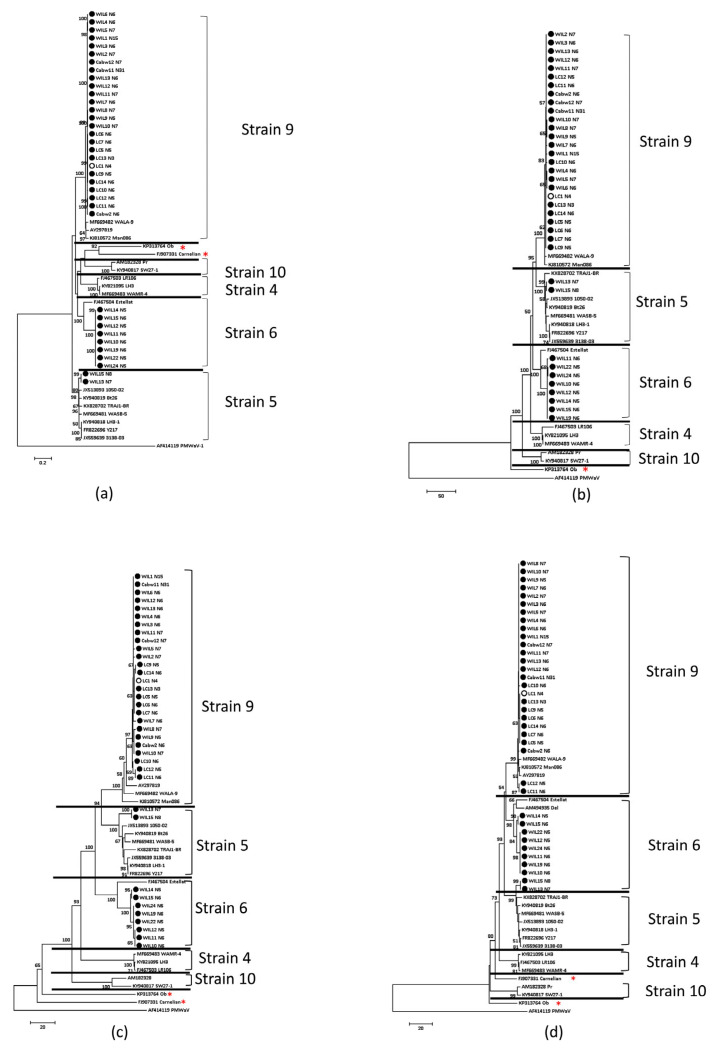
Neighbour-joining phylogenetic trees of grapevine leafroll-associated virus 4 (GLRaV-4) detected by metagenomic high-throughput sequencing and isolates from the GenBank database. Phylogenetic trees constructed by alignments of (**a**) complete genome sequences (above 12600 nts), (**b**) amino acid (aa) sequences of the RNA-dependent RNA polymerase (RdRp), (**c**) aa sequences of the heat shock protein 70 homologue (HSP70h) and (**d**) aa sequences of the coat protein (CP) gene. Bootstrap values less than 50% are not shown. * Indicates GLRaV-4 isolates with similarities to the exemplar isolate LR106 (accession no. FJ467503) below the demarcation of this species. Open circles indicate sequences generated by a previous study [1] and black dots display the sequences by this study.

**Table 1 viruses-15-00774-t001:** The number of grapevines, their location, variety/clone, year planted and symptoms selected for the reverse transcription polymerase chain reaction (RT-PCR) and metagenomics high-throughput sequencing (Meta-HTS) experiments.

Location	Year Selected	Variety and Clone	Year Established	Grafted on Rootstock?	Total Grapevines		No. of SD Grapevines		No. of LRD Grapevines		No. of Asymptomatic Grapevines	
					RT-PCR	HTS	RT-PCR	HTS	RT-PCR	HTS	RT-PCR	HTS
Langhorne Creek ^1^ (LC)	2018	Shiraz BVRC12	2004	Chardonnay	30	14	15	9	N/A	0	15	5
Willunga (WIL)	2018&2020	Shiraz BVRC12	2004	No	80 ^4^	24	8	13	7	7	15	4
	2020	Cabernet Sauvignon SA125 ^2^	2004	No	4	4	N/A	N/A	4	4	N/A	N/A
	2020	Merlot unknown clone ^2^	Unknown	Unknown	1	1	1	1	N/A	N/A	N/A	N/A
Barossa Valley ^3^ (BV)	2018	heritage Shiraz, unknown clone	≈1900σ	No	6	3	N/A	N/A	3	2	3	1
Coombe vineyard (CV)	2020	Shiraz ^2^ BVRC12	1993	No	2	2	N/A	N/A	N/A	N/A	2	2
	2020	Cabernet Sauvignon SA125 ^2^	1993	No	2	2	N/A	N/A	2	2	N/A	N/A

^1^ No grapevine with leafroll disease symptoms was found at the LC site. ^2^ For studying the infection source at the WIL site only. ^3^ BV grapevines showed mild leafroll disease symptoms. ^4^ Thirty LC grapevines were chosen in 2018 but were subsequently removed, therefore 50 additional WIL grapevines were sampled in 2020 for the analysis.

**Table 2 viruses-15-00774-t002:** The presence of grapevine virus A (GVA), and grapevine leafroll-associated viruses 3 (GLRaV-3) and -4 (GLRaV-4) determined by endpoint RT-PCR and metagenomic high-throughput sequencing (Meta-HTS), and associated symptoms, in 50 grapevines.

Sample ID ^1^	Total Number of Grapevines with Symptom and Virus Status Combination	Symptoms ^2^	GVA (I,II,II), GLRaV-1 (1), GLRaV-3 (3) and GLRaV-4 (4) Status by RT-PCR ^3^	GVA (I,II,III), GLRaV-1 (1), GLRaV-3 (3) and GLRaV-4 (4) Status by Meta-HTS ^4^
WIL19, WIL22	2	Asymptomatic	4	4
WIL17, LC16, LC18, LC20, LC24, LC27, BV6, CVP5, CVP6	9		None	None
WIL24	1		None	4
WIL14, 15	2	LRD	3, 4	3, 4
WIL9, WIL10, WIL11, WIL12	4		III, 3, 4	III, 3, 4
Cabw1, Cabw2, Cabw12	3		I/II, III, 3, 4	II and III, 3, 4
Cabw11	1		I/II, 3, 4	II, 3, 4
WIL13 *	1		3, 4	III, 3, 4
BV1, 3	2	Mild LRD	I/II, 1	I, 1
CabSA125_R3V30, CabSA125_R3V44	2		I/II, III, 4	II and III, 4
WIL8	1	SD	I/II, III, 3, 4	II and III, 3, 4
LC10, LC11, LC12, LC13, LC14	5		I/II, 4	II, 4
WIL48, WIL49, WIL50, WIL53	4		1/II, 3	II, 3
WIL4, WIL5, WIL6, WIL7	4		I/II, 3, 4	II, 3, 4
LC5, LC6, LC7, LC9 *	4		I/II	II, 4
WIL47 *	1		I/II, 3	III, 3
Melort1 *	1		I/II, III, 3	II and III, 3, 4
WIL1, WIL2, WIL3 *	3		I/II, III, 3, 4	II andIII, 4

^1^ WIL = Willunga, LC = Langhorne Creek, BV = Barossa Valley, CV = Coombe’s Vineyard. All grapevines listed are var. Shiraz except Cabw = Cabernet Sauvignon from Willunga and CabSA125 = Cabernet Sauvignon clone SA125 from Coombe’s Vineyard. ^2^ I = GVA^I^, II = GVA^II^, I/II = GVA^I^/GVA^II^ (the two groups cannot be discriminated by RT-PCR), III = GVA^III^, 1 = GLRaV-1, 3 = GLRaV-3, 4 = GLRaV-4. ^3^ SD = Shiraz disease, LRD = leafroll disease. ^4^ GVA^I^, GVA^II^ and GVA^III^ phylogenetic groups were identified using contigs generated by Meta-HTS (Appendix A). * RT-PCR and Meta-HTS results mismatched.

**Table 3 viruses-15-00774-t003:** Sequence similarity within and between phylogroups of grapevine virus A (GVA) of the Australian isolates obtained by metagenomic high-throughput sequencing and international isolates from the GenBank database.

Genes Compared ^1^		I ^2^		II ^2^		III ^2^	
		Australian Isolates Only	All Isolates	Australian Isolates Only	All Isolates	Australian Isolates Only	All Isolates
Whole genome		N/A ^3^	76.33–99.80	91.45–99.90	79.92–99.90	94.85–99.94	76.52–99.94
RdRp	RdRp nt	N/A	74.96–99.88	90.65–99.90	79.53–99.90	94.12–99.96	74.76–99.96
	RdRp aa	N/A	84.72–99.53	96.60–100	90.28–100	97.25–100	86.59–100
MP	MP nt	99.88 ^4^	77.30–100	91.07–100	83.03–100	96.31–100	81.67–100
	MP aa	100 ^4^	78.85–100	92.47–100	89.25–100	95.70–100	87.46–100
CP	CP nt	98.49–99.83	80.23–99.66	94.30–100	84.09–100	97.82–100	77.39–100
	CP aa	98.99–100	79.90–100	96.98–100	88.94–100	98.99–100	82.41–100
RB	RB nt	95.24–100 ^5^	86.08–100 ^5^	See GVA^I^		97.43–100	91.58–100
	RB aa	94.51–100 ^5^	84.62–100 ^5^			97.80–100	92.31–100

^1^ RdRp = RNA-dependent RNA polymerase, MP = movement protein, CP = coat protein, RB = RNA-binding protein, nt = percentage nucleotide identity, aa = percentage amino acid similarity. ^2^ I = GVA^I^, II = GVA^II^, III = GVA^III^, grapevine virus A isolates from the phylogroups I, II and III. ^3^ Only one whole genome sequence from group I was available from Australia. ^4^ Only two sequences were compared. ^5^ Identities or similarities obtained from both GVA^I^ and GVA^II^ isolates.

## Data Availability

The sequencing data were submitted to the GenBank database and can be accessed using the accession numbers provided at the National Center of Biotechnology Information website [61].

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
