# Peer review of "A Metagenomic Investigation of the Viruses Associated with Shiraz Disease in Australia"

_viruses, 2023, doi:10.3390/v15030774_

Round 1

Reviewer 1 Report

The “Virome study of two Australian Shiraz Disease vineyards” is the subject of the manuscript of Wu and colleagues. Shiraz Disease is a very curious disease as it occurs in Australia and South Africa and on a limited number of cultivars. The authors takes the cue from the existing indications of Grapevine virus A involvement in SD to perform a thorough investigation of the virome of vines from three vineyards, considering also other major grapevine viruses. The data support the involvement of GVA variants II in the genesis of SD. The work represents an additional information to the etiology of SD but it is not resolutive. In my opinion, since grapevine is affected by multiple viruses and many of them exist in different variants, a real progress in such investigations need functional genomic studies, as also suggested by the authors. The manuscript does not present major criticisms either in the language or in the methodology and therefore can be accepted for publication, providing solving minor comments as below reported:

Row 92, the adjoining vineyards  (CS and Merlot) have been  described poorly as sort of byproduct with a reduced  number of RT-PCR tests : this tests mainly in the borders along the test Shiraz vines could better explain if the virus spread has a directional propagation ( from old to new plantation ?)

Row 183. De novo should be italic

Row 203, the best fit substitution model is missing in the choice of the pilo tree design, given the quantity of isolates (which could be merged or from which a few references could be selected per cluster) a maximum likelihood would be more precise (but perhapsmore expensive)

Row 241, It was surprising to me the lack of GFKV and GVB. How do you explain it?

Rows 611-613. This sentence is quite obvious. I suggest to eliminate it.

The paper is full of Tables. Perhaps Table S4 and S5 could be merged

Author Response

Q1: Initially, the author panel discussed whether to test neighboring Merlot and Cabernet for viruses to see if they are infected or not with viruses associated with SD. We noted that the data from a few of these neighboring grapevines was insufficient to find out the directional propagation. However, it is important to show the potential of spread of the associated viruses between blocks, which could have led to SD at WIL, or LRD and SD in the CS and Merlot1 blocks. If we want to investigate further to see if the virus spread has a directional propagation, certainly more RT-PCR and Meta-HTS analysis would be needed on CS and Merlot. Consequently we have reworded the Sentence at line 94  as follows “To investigate the potential for spread between WIL and adjoining blocks of grapevines,….”.

Q2: Corrected.

Q3: We have decided to use the neighbour-joining method, which does not use a best fit substitution model, because the previous GVA phylogenetic analysis by Goszczynski from South African [11] was done using a neighbour-joining method and we wanted to compare to see if our data support the previous grouping system. The following line has been added to section 2.5.3 to reflect this: “The neighbour-joining method was used to enable comparison of GVA phylogenetic groupings reported in previous studies [11].” Q4: We believe it is not surprising that GFKV or GVB were not detected in our samples. Australia has several grapevine vine improvement programs that enable access to material that has been pathogen tested and in which viruses, other than GRSPaV, are not detected; this can limit prevalence of some viruses such as GFkV which do not have a known vector. It is likely the vineyard was planted with high health material. Diagnostic testing over many years indicates that GVB occurs with low prevalence in Australia. This is due to control measure placed at the Australian border to prevent introduction of corky bark disease, which is associated with strains of GVB, and is not present in Australia.   Q5: Deleted Q6: We also agree that there are common headings in both S4 and S5, but they are different in content therefore it is important to keep both tables separate.  

Reviewer 2 Report

The manuscript presented by Wu et al. used RT-PCR and metagenomic high throughput sequencing to study the virome of symptomatic and asymptomatic grapevines within vineyards affected by SD and located in South Australia. The results showed that grapevine virus A (GVA) phylogroup II variants were strongly associated with SD symptoms in Shiraz grapevines, while GVA phylogroup III variants and GVA phylogroup I variants may not be associated with SD. The title of "Virome study of two Australian Shiraz Disease vineyards" only emphasizes the "Virome study", but does not reflect this main contribution. It is suggested that the author revise the title. Generally, the manuscript presented and well analyzed quite a lot of data and written well. I would like to offer the following suggestions with the aim to improve the manuscript.

Comments:

1.     The full name of the virus should be used instead of abbreviation when it first appears, such as Line 161 "GRSPaV" and L241 "GRVFV".

2.     2. It is recommended to use Figure S2 as Figure 1 in the text and add asymptomatic pictures. And it is recommended to provide symptom photos of single virus infection and multiple virus co-infection.

3.     Why do the authors use different methods to extract RNA for RT-PCR and deep sequencing? Does it affect the detection results of viruses?

4.     Is there any difference in the concentration of dsRNA extracted from asymptomatic samples and symptomatic samples?

5.     L282-283,The detection of GVA by Meta-HTS generally corresponded with the results of the 282 endpoint RT-PCR assays with some exceptions.

6.     Please check whether the resolution of the picture meets the published requirements.

7.     Is the "sequence similarity analysis" in Table 3 based on Sanger sequencing or depth sequencing?

8.     It is recommended to provide more results of virome analysis, such as the abundance of each virus in the sample.

Author Response

New title: “A metagenomic investigation of the viruses associated with Shiraz disease in Australia.”

Q1: Added the full name.

Q2: Figure S2 has been moved to Figure 1 as suggested. In the four vineyards studied, there were no grapevines that were infected with only GVA, but we have included the phylogenetic groups of GVA variants at each location in the figure caption. We added a photo of asymptomatic Shiraz BVRC12 as Figure 1d.

Q3: Initial testing of grapevines was done by RT-PCR and a more rapid total RNA extraction method, compared to the dsRNA extraction method, was used. Our initial HTS experiment compared two extraction methods using the same samples on a MiSeq instrument. The data suggested that samples prepared by the dsRNA method generated significantly higher number of viral reads and assembled more viral contigs than the TNA extraction method used in the RT-PCR detection. We did RT-PCR testing on dsRNA samples prior to library preparation to ensure they have the same virus status as the TNA method.

Q4: Prior to ribo-depletion, we adjusted concentration of total RNA in each sample to the same.

Q5: We found that Meta-HTS matched with RT-PCR, but in some samples, Meta-HTS detected more viruses than RT-PCR or vice versa.

Q6: Will provide higher resolution photos.

Q7: Added to Table 3 heading “isolates obtained by metagenomic high throughput sequencing”.

Q8: We provided the frequency of each virus in total samples in section 3.1. If you mean the proportion of viral reads of each virus in each sample, it is out of the scope of this paper.